# Development and Application of Healthiness Indicators for Commercial Establishments That Sell Foods for Immediate Consumption

**DOI:** 10.3390/foods10061434

**Published:** 2021-06-21

**Authors:** Letícia Ferreira Tavares, Patrícia Maria Périco Perez, Maria Eliza Assis dos Passos, Paulo Cesar Pereira de Castro Junior, Amanda da Silva Franco, Letícia de Oliveira Cardoso, Inês Rugani Ribeiro de Castro

**Affiliations:** 1Institute of Nutrition Josué de Castro, Federal University of Rio de Janeiro, Rio de Janeiro 21941-902, Brazil; leticiatavares@nutricao.ufrj.br (L.F.T.); elizapassos@nutricao.ufrj.br (M.E.A.d.P.); paulocastro@nutricao.ufrj.br (P.C.P.d.C.J.); 2Institute of Nutrition, State University of Rio de Janeiro, Rio de Janeiro 20559-900, Brazil; patricia.perez@uerj.br (P.M.P.P.); inesrrc@uol.com.br (I.R.R.d.C.); 3Nutrition Coordination, Health Sciences Center, Centro Universitário Serra dos Órgãos (UNIFESO), Rio de Janeiro 25964-000, Brazil; franco.amandarj@gmail.com; 4Oswaldo Cruz Foundation (FIOCRUZ), National School of Public Health, Rio de Janeiro 21041-210, Brazil

**Keywords:** consumer food environment, assessment, ultra-processed foods

## Abstract

Studies of food environments lack easy-to-apply indicators for their characterization and monitoring. This study aimed to create and assess the applicability of an a priori classification of establishments that sell foods for immediate consumption and to develop and apply indicators for assessment of the establishments’ healthiness. The indicators were grouped by the types of foods sold most frequently at these establishments, according to the extent and purpose of the foods’ industrial processing. Four indicators were developed, based on the availability of unprocessed/minimally processed foods (MPF) and ultra-processed foods (UPF) in the establishments. The classification and indicators were applied to commercial food establishments at two Brazilian universities. Descriptive analyses were performed to characterize the food environment for all the establishments and by university. Two proportion indicators assess the relative availability of subgroups of MPF and UPF. The UPF/MPF ratio expresses the relative advantage/disadvantage of the availability of MPF compared to that of UPF. The Healthiness Index or summary score expresses the availability of MPF and the unavailability of UPF. The classification and indicators present good discriminatory power and are easy to operationalize, interpret, and adapt.

## 1. Introduction

The food environment can be defined as “the physical, economic, political and socio-cultural context in which consumers engage with the food system to acquire, prepare and consume food” [1]. Different conceptual models for characterizing the food environment have been proposed in recent decades [2,3,4,5,6]. Among them, the model proposed by Glanz et al. (2005) [3] has been widely adopted in studies that seek to characterize food environments in different settings [7,8] and examine their association with health outcomes [9,10,11]. One of the types included in this model is the “consumer food environment”, demarcated by what is found within and around establishments that sell (or supply) foods, beverages, preparations, and convenience items. The establishments can either be located within territories, comprising the “community food environment”, or available to specific groups, as in workplaces, schools, and universities, constituting the so-called “organizational food environment” [3].

Although the major body of the literature is focused on the characterization of food retail from the perspective of the “community food environment”, with a focus on availability and physical accessibility of establishments that sell food, it is important to complement this approach with the study of the “consumer food environment”, since this allows characterizing the products effectively sold, their price and marketing strategies that, as a whole, influence/determine food choices [3].

In recent years, much progress has been made in relation to the development and application of instruments to assess different scenarios in the food environment [12,13,14]. In relation to the “consumer food environment”, studies often present descriptive results of the respective food environment [15,16,17,18]. However, the results of these studies are usually presented item by item and, although the use of summary indicators is accepted as an important step for characterizing food establishments [19], studies focused on developing such indicators are still incipient.

There are still few studies that present summary measures for the assessment of the consumer food environment [13]. Most summary indicators proposed in these studies are only applicable in certain types of establishments that sell foods to be consumed later, such as supermarkets [20,21,22,23,24]. Besides, the construction of these measures is quite complex [21,23,25], which hinders their large-scale use and/or application to specific realities [18]. 

The lack of simple, reliable, and valid summary indicators for characterizing the consumer food environment in establishments that sell foods for immediate consumption limits the interpretation of results from increasingly frequent initiatives to characterize and monitor these settings. The development of such summary indicators would allow comparison of studies within the same country and between countries, besides favoring the study of relations between the consumer food environment and food purchases or food consumption behaviors [21,26,27]. The application of such summary indicators can also assist institutional diagnoses (in schools, universities, hospitals, and other workplaces) and community diagnoses at different levels (neighborhood, city, and even country), besides monitoring these environments over time. These applications could contribute to the development and evaluation of interventions and public policies aimed at improving food environments to make them healthier [19,26].

Based on the above, this study’s objectives were to create and assess the applicability of an *a priori* classification of establishments that sell foods for immediate consumption (snack bars, cafeterias, bars, restaurants, and candy stores), to develop summary indicators to assess the healthiness of these establishments and apply them to the environment in two public universities in the city of Rio de Janeiro, Brazil. We chose universities for application of the proposed indicators because they are complex organizational environments and have a wide range of establishments that sell foods for immediate consumption, thus providing an appropriate scenario for this analysis [28]. 

The contribution of this study is to offer simple and valid summary indicators for characterizing the consumer food environment in establishments that sell foods for immediate consumption.

## 2. Materials and Methods

### 2.1. Data Collection Instrument

The data collection instrument that provided the basis for the elaboration of the indicators proposed here was a checklist applied with the audit method. An audit is the method most frequently reported in the literature for measuring the consumer food environment [12]. It is used to assess the variety of products, nutritional information, prices, advertising, and other aspects pertaining to the foods’ availability that can characterize the establishments as facilitators or barriers to healthy choices in the foods’ acquisition [29]. Across different audit tools, the checklist has been the most widely used [12]. It includes a limited list of items that are selected according to predetermined criteria. It was chosen for the current study because it allows characterizing the establishments and constructing simplified indicators for assessment of the consumer food environment. 

The instrument was based on the theoretical approaches in Glanz et al. (2005) [3] and Caspi et al. (2012) [29]. From Glanz et al. (2005), based on the theoretical category “consumer food environment”, defined in the introductory section, the instrument included elements referring to “nutritional information” and “food promotion”. From Caspi et al. (2012), the instrument included elements referring to the dimensions “availability”, “affordability”, and “convenience (accommodation)”. 

The checklist is structured in seven different domains, namely: characterization of the establishment; observation of the environment; general information furnished by the establishments to consumers (nutritional information, prices, and menu); prices and promotions; advertising; availability of beverages, foods, and preparations; and availability of convenience items. The latter two domains, referring to availability, include foods for immediate consumption that were selected according to evidence that high consumption was associated with protection against (or risk of) chronic noncommunicable diseases [30,31] (questionnaire available as Appendix A). The reliability and content validity of the instrument were evaluated by Franco (2016) [15].

For the current study, we selected the domains pertaining to characterization of the establishment and availability of foods for immediate consumption, since our interest was to develop indicators based on the availability of foods. We chose to focus on indicators of availability since this was the fundamental dimension of the access construct [29] and the easiest to measure; thus, indicators based on this dimension provide essential information on the food environment and are the easiest to operationalize, which we consider a key aspect for the current study’s development. Given its importance and the greater ease in its measurement, it is the most frequently evaluated dimension in studies on consumer food environment. Therefore, the development of summary measures built exclusively on availability data can contribute to improving the reinterpretation of data already collected in different realities.

### 2.2. Classification of Foods

The foods were first categorized in two groups, ultra-processed and others (unprocessed, minimally processed, or processed), considering the extent and purpose of their industrial processing, as proposed by the NOVA classification [32]. These categories are also used in the recommendations of the Dietary Guidelines for the Brazilian Population [33]. They were organized in subgroups, and the classification’s preliminary version was analyzed by five researchers from the field of nutrition and affiliated with different institutions, with experience in the issue of food environment. The classification’s final version was thus structured as groups of unprocessed/minimally processed foods (MPF) which included nine subgroups: (i) raw vegetables, (ii) cooked vegetables, (iii) fresh fruits, (iv) natural fruit juice, (v) whole grain rice, (vi) legumes, (vii) homemade dressings (viii) coconut water and water, and (ix) coffee.

The group of ultra-processed foods (UPF) also included nine subgroups: (i) ultra-processed snacks (cereal bars; breakfast cereal; whole grain cookies; cream-filled cookies; cookies without filling; packaged savory snacks or crackers without filling), (ii) savory snacks (fried/baked), (iii) ultra-processed sauces (readymade dressings; shoyu), (iv) confectionery (bonbons and chocolate bars; candy), (v) sweet culinary preparations containing processed/ultra-processed foods (eclairs, cupcakes, cakes, pies), (vi) sodas (including flavored water), (vii) other sugar-sweetened beverages (ready-to-drink tea; guarana drink; industrialized beverages made of fruit juices or fruit nectar; soy-based industrialized beverages; flavored milk or milk-based beverages or industrialized yogurt), (viii) energy and sports drinks, and (ix) industrialized fruit drinks. The food items’ allocation to the subgroups was based on nutritional similarity and purpose of consumption.

To explore the selected subgroups’ capacity to explain the total variability of the MPF and UPF groups, we performed factor analysis and analysis of internal consistency [12,34,35,36]. Confirmatory factor analysis with varimax rotation was conducted using the 18 subgroups. The analysis was carried out with a number of fixed factors, because the objective was to classify establishments according to the owners in two food groups (MPF and UPF). In the preliminary stage of analysis, we used the scree graph and three factors would be retained (MPF, UPF, and beverages). Two factors were retained, since the factor loadings remained adequate, and having a group with only beverages would not be recommended, since the latter are available in large proportions of the establishments (varying from 83.3% to 98.7%, respectively, for industrialized juice and water). Internal consistency of the subgroups’ classification in the two groups of foods was assessed with Cronbach’s alpha, assuming 0.70 as the cutoff point [36].

The results of the Kaiser–Meyer–Olkin (KMO) test (0.648) and Bartlett’s sphericity test (*p*-value < 0.001) showed that the data assumptions were adequate for the factor model’s application. Total accumulated variance was 35.4%, and two patterns were identified that represented the two predefined groups of foods: 1. MPF with 10 items (classified *a priori* and industrialized fruit refreshments). In the first extracted factor, only two of the items showed factor loads less than 0.3 (coffee 0.25 and industrialized fruit refreshments 0.24) and high internal consistency (Cronbach’s alpha 0.718); The second had a UPF with eight items, all with adequate factor loads (>0.3) also with high internal consistency (Cronbach’s alpha 0.737—data not shown). These results suggest an adequate conformation of the groups. 

### 2.3. Classification of the Establishments

The checklist categorizes the establishments that sell foods for immediate consumption in eight types according to the predominant activity. This classification was based on the most common types of establishments found in Brazil and that applied to the reality of the points of sale in universities, the organizational food environment for which the instrument was developed. They are: (i) buffet restaurants by weight (meals and food items by weight); (ii) à la carte/fixed dish restaurants (meals with fixed amounts and prices); (iii) all-you-can-eat buffet restaurants, fixed price; (iv) candy stores (candy, sweets, and sugar-sweetened beverages); (v) snack bars (quick snacks and candy items); (vi) bars (alcoholic and nonalcoholic beverages, quick snacks, meals, and candy); (vii) cafeterias (coffee and nonalcoholic beverages, and, sometimes, full meals); (viii) mixed (snack bars or cafeterias and buffet by weight or la carte/fixed dish restaurants).

Having identified the types, as in other studies [9,37], the establishments were grouped *a priori* based on the types of foods sold most often in these establishments, according to the degree and purpose of the industrial processing, using the NOVA classification [32], namely: type 1: predominantly MPF (buffet restaurant by weight; à la carte/fixed dish restaurant; and all-you-can-eat buffet); type 2: mixed (no predominance of MPF or UPF), (snack bar/cafeteria with the sale of meals); and type 3: predominance of UPF (snack bar/cafeteria without sale of full meals, candy store, and bar). The five researchers consulted on the food groups also gave their opinions on this *a priori* classification of the establishments. 

### 2.4. Development of Indicators

Four indicators were developed for the availability of MPF and UPF in the establishments (Table 1). Each indicator was described as follows: indicator/acronym (name that identifies the indicator succinctly and clearly and its respective abbreviation); formula for calculation (represents the method for the indicator’s calculation, based on its constituent variables); purpose (objective of measurement using the indicator); and example (based on hypothetical data, presents the calculation of the indicator and the result’s interpretation) [38,39].

Of the 32 food items that comprised the subgroups, only three (homemade dressings, readymade dressings, and shoyu) had missing data, and they were imputed in nine (11.5%) of the 78 target establishments. The response observed most frequently for each item in the establishments of the same type was imputed for the respective missing data. We observed similarity in the variables’ distribution before and after imputation. 

Two indicators of proportion were elaborated: the first expresses the relative availability of each subgroup of MPF in relation to the total for all the selected subgroups of MPF (Prop-MPF). The second expresses the relative availability of each subgroup of UPF in relation to the total for all the selected subgroups of UPF (Prop-UPF). 

The third indicator is the ratio between the availability of subgroups of these two groups (Ratio-UPF/MPF). The objective is to express whether the food establishment sells more subgroups of UPF than subgroups of MPF. That is, it allows identifying the relative advantage/disadvantage in the availability of MPF compared to that of UPF: if the number of available subgroups of UPF is greater than that of subgroups of MPF, this indicator will be greater than 1; if the number of available subgroups of MPF is greater than that of subgroups of UPF, the indicator will be less than 1. Application of the indicator allows assessing the “competition” between the two food groups, MPF and UPF. This indicator has a mathematical limitation, because the calculation cannot be performed if the denominator is equal to zero. 

The fourth indicator is the Healthiness Index (HI), a summary measure of the availability of MPF and UPF. It is intended to express the idea that an establishment that promotes healthy eating is one that both facilitates access to MPF and hinders access to UPF [26,33]. From this perspective, the indicator was built as follows: the establishment wins a point for each of the nine subgroups of MPF sold and for each of the nine subgroups of UPF not sold; for the other situations, the establishment receives no points. The establishment thus receives the maximum score if it sells all nine subgroups of MPF and none of the nine subgroups of UPF. To facilitate the score’s interpretation, we converted the score (which can vary from 0 to 18) to a scale from 0 to 100. It should provide a concise and precise description of the experimental results, their interpretation, as well as the experimental conclusions that can be drawn.

Figure 1 summarizes our methodological approach, from the theoretical references to the elaboration of the proposed indicators.

### 2.5. Application of the Indicators

Calculation of the indicators was applied to data collected from commercial food establishments in two public universities in the city of Rio de Janeiro, Brazil. In the State University of Rio de Janeiro (UERJ), the sample included the main campus, which is a 12-story vertical urban campus with 24 academic units, with daily circulation of about 35,000 people (faculty, technical and administrative staff, students, and the floating population). All 25 establishments on this campus participated in the study. In the Federal University of Rio de Janeiro (UFRJ), the sample also included the main campus. This is a horizontal campus with 26 academic units and two healthcare units, with a mean daily circulation of 65,000 people. Of the 57 existing establishments, 53 agreed to participate. The audit of all the establishments that sold foods in the universities was conducted from November 2015 to February 2016 with an instrument previously developed, by previously trained evaluators. 

### 2.6. Data Analysis 

The applicability of the a priori classification was examined with the indicators proposed for the establishments in each of its three types. The resulting mean values were compared, and statistical significance of the differences was examined via confidence intervals.

The proposed indicators were used to perform descriptive analyses for characterization of the food environment, based on the establishments as a whole and by university. Comparison of the indicators between the two universities was based on the difference of the summary measures and analysis of the 95% confidence intervals. 

Data entry by two independent individuals and validation of the dual data entry were performed with Microsoft Excel^®^ 2007. Data analysis used the Statistical Package for the Social Sciences (SPSS) version 21.0 [40]. 

The study was approved by the Institutional Review Board of Hospital Pedro Ernesto, State University of Rio de Janeiro (approval number CAAE 49988015.6.0000.5259).

## 3. Results

When the indicators were applied to the universities, UFRJ had 53 establishments selling foods for immediate consumption with different profiles (n = 21 mixed, 15 snack bars, 9 à la carte or fixed-dish restaurants, 4 buffet restaurants by weight, 3 candy stores, and 1 cafeteria). The UERJ had 25 establishments (n = 13 snack bars, 9 mixed, 2 candy stores, and 1 à la carte/fixed-dish restaurant). Neither of the universities had a bar or an all-you-can-eat buffet restaurant. 

The adequacy of the *a priori* classification was confirmed by the clear gradient between the three types of establishments for the indicators Prop-MPF, Ratio-UPF/MPF, and HI and between the first and the other types of establishments for the indicator Prop-UPF, always with better results for Group 1 in relation to the availability of MPF and unavailability of UPF (Table 2). Except for the results obtained with the indicator Prop-UPF in establishments types 2 and 3—the observed differences were statistically significant.

Considering that the indicators can be used to assess either one establishment or a set of establishments, we conducted the exercise of consolidating the results from all the establishments at each university and comparing the institutions (Table 3). The consolidated results by university allow knowing the overall profile and the variability of the set of establishments. We found that the indicators were able to discriminate between two different profiles of food environments in similar contexts (public universities in this case). The distribution of types of establishment are consistent with the frequencies of calculated indicators: worse results were observed in the university with a higher proportion of type 3 establishments, and better results in the university with a higher proportion of type 1 establishments.

## 4. Discussion

This study presented the development and application of an *a priori* classification of types of establishments that sell foods for immediate consumption and four indicators of availability of foods in these establishments. The classification and indicators showed good discriminatory power in the context in which they were applied. This finding on the *a priori* classification suggests that it can be useful for characterizing, monitoring, and assessing the food environment in settings where it is not possible to conduct audits to check the presence of MPF and UPF subgroups with a view towards building the proposed indicators. 

The indicators based on the NOVA food classification [32] and presented here were able to concisely portray the availability dimension in the food environment. The four indicators complement each other in the characterization of the foods’ availability in an establishment (or set of establishments) and indicate, from different perspectives, the degree to which the establishment favors or hinders healthy food choices. The Prop-MPF and Prop-UPF indicators reveal the relative availability of specific MPF and UPF subgroups in relation to the respective total of subgroups, indicating the degree to which the establishment approaches a healthy food environment for each of the food groups (selling few or no subgroups of UPF and/or many subgroups of MPF) or differs from a healthy food environment (selling many subgroups of UPF and/or selling few subgroups of MPF). Ratio-UPF/MPF expresses the relative advantage/disadvantage in the availability of MPF in relation to UPF, that is, whether there is a predominance of the availability of UPF subgroups over MPF subgroups or vice versa, i.e., consistent with the notion of “competition” between the food groups. Meanwhile, the Healthiness Index (HI) summarizes the combined availability of MPF and UPF, offering an overall assessment of the establishment (or set of establishments) through a score that measures the availability of MPF and the unavailability of UPF. 

The proposed indicators are easy to interpret and dispense with a detailed audit of each establishment, thus simplifying data collection when constructing them. They can be useful for characterizing an establishment (or set of establishments) at a given moment, monitoring it over time, establishing targets to improve its routine operations, assessing interventions to improve the food environment, and comparing establishments or sets of establishments in these four applications (characterization, monitoring, definition of targets, and assessment of interventions). These indicators can be applied in any establishment that sells food for immediate consumption, regardless where it is located: workplaces, schools, universities, hospitals, shopping centers, metro stations, communities, neighborhoods, etc. 

The indicators’ practicality favors their use not only in academic studies, but also in public policy initiatives to improve the food environment and in the establishments’ routine management. For example, establishment managers, motivated to make their food environment healthier, can evaluate food availability, set goals for its improvement, develop strategies for implementing these improvements, and regularly apply the indicators proposed here to monitor the achievement of goals. For universities, schools, workplaces, etc., beyond these applications, the indicators can guide the hiring of food services which will work on their premises.

Depending on the context and objective of their application, one can apply all or some of the indicators. Our indicators can also be adapted, using different subgroups of MPF and UPF from those adopted here, while maintaining the logic of relative shares of subgroups of foods (indicators of proportion), relative disadvantage/competition between the subgroups (availability ratio), and summary indicator (Healthiness Index). This adaptation may be useful in the future for maintaining the indicators’ discriminatory power, given the market dynamism in creating and offering new foods in establishments like those analyzed here.

Note that the indicators were conceived for establishments that offer foods for immediate consumption. Although these are the most common types in organizational food environments (e.g., workplaces, schools, universities, hospitals, etc.), these establishments represent only part of the supply in community food environments (territories, neighborhoods), which also include establishments that sell foods for preparation and consumption later in time (e.g., supermarkets, butcher shops, and open-air or farmers’ markets) [3]. Thus, for a complete assessment of the establishments in community food environments, it would be necessary to adapt and/or complement the indicators proposed.

Among the few studies that present summary measures, the complete or adapted tool of the Nutritional Environment Measures Survey (NEMS) has been the most widely used in studies to assess the consumer food environment [22,25,41,42,43]. NEMS uses a composite indicator that assesses the availability of healthy food choices, prices, and quality of foods, attributing points to the healthy food options based on a combination of the three dimensions analyzed. There are versions of the instrument both for the assessment of the retail food trade and the assessment of establishments that sell foods for immediate consumption. The complexity of the tool’s implementation, the subjectivity of the quality assessment, and the proposal to assess only the presence of healthy food options (overlooking the presence of unhealthy options) have been some of the barriers to the use of different versions of the NEMS [22,41].

Duran et al. (2013) [42] developed the Healthy Food Store Index (HFSI), an adaptation of the NEMS-S for assessment of the retail food trade in the Brazilian context that evaluates the availability, variety, and advertising of healthy foods (fruits and vegetables) and some ultra-processed foods (sugar-sweetened beverages, savory corn snacks, and cream-filled cookies). Although HFSI has been used for the assessment of the retail food trade in different contexts [42,44], it has the same limitations as the NEMS and only allows assessing a small variety of foods. 

Other instruments and indicators have been adopted to measure the consumer food environment with a focus on the assessment of facilitators for the consumption of healthy foods, with availability, prices, and information of healthy foods in establishments such as convenience stores, markets, and grocery stores [45,46,47]. More recently, an audit tool based on the NOVA classification (AUDITNOVA) was developed for the assessment of the retail food trade [21,23]. All the above-mentioned instruments were developed to characterize the retail food establishments (supermarkets, grocery stores) and they are, thus, not adequate for the assessment of establishments that sell foods for immediate consumption. 

For the assessment of the consumer food environment in universities in New Zealand, Roy et al. [48] used a healthiness index based on the New Zealand food and nutritional guidelines and front-of-package labeling and health star rating system. Unlike the index presented here, which assessed the dimension of “availability” in the food access construct [29], the indicator proposed by the authors also considered the dimensions “physical accessibility” and “affordability”, besides the existence of promotions in the establishments. Although the index used by Roy et al. [48] assesses more than one dimension of food access, its implementation is more complex, since it takes into consideration the comparison of the foods’ costs, based on the average price practiced in the country’s principal supermarkets. In addition, since it was formulated according to specific guidelines in New Zealand, the indicator has limited application in other realities. Furthermore, an advantage of the set of indicators presented here in comparison to that proposed by Roy et al. [48] is the possibility of identifying competition between the available groups of foods in the commercial establishments and allowing an overall assessment that enables the determination of a given food environment’s healthiness.

In addition, the indicators developed in the current study are the first to consider the NOVA classification of foods [32] for assessment of the consumer food environment through verification of the dimension of availability of foods for immediate consumption in establishments such as restaurants, snack bars, and others that comprise the food environment. 

Our study has some limitations. The fact that the indicators are built with variables based on the presence (or absence) of food subgroups without considering the amounts and variety of food items in each subgroup prevents a more detailed characterization of the establishments and may potentially underestimate (for example) the availability and predominance of UPF. By way of illustration: for the indicators’ construction, in relation to this subgroup of UPF, one establishment that sells 10 different types (varieties) of cookies without filling, packaged savory snacks or crackers without filling is treated the same way as another establishment that sells 50 varieties of these foods. However, we opted to propose indicators based on a qualitative logic (presence or absence of food subgroups rather than counting types/varieties), aimed at facilitating its application. The way the indicators are constructed, their application does not require highly trained personnel, which is an advantage for their operationalization. Complementary indicators to those presented here can be developed to resolve the above-mentioned limitation.

In addition, the proposed indicators only assess the foods’ availability, without considering the other four dimensions of the food access measure as described by Caspi et al. [29], namely: physical accessibility, affordability, acceptability, and accommodation. Although availability is an essential dimension for the others to be analyzed, researchers interested in investigating these other dimensions will need to complement their assessment with additional indicators besides those proposed here. 

The fact that the databases used here refer to the scenario in 2015/2016 in the respective universities is a limitation for the evaluation of the current food environment scenario of the studied universities. However, until the beginning of the COVID-19 pandemic, when the universities interrupted their in-person activities, the profile of the food environment in the universities studied was similar to that recorded in 2015/2016. Therefore, since our objective was to examine the indicators’ performance rather than describe the universities’ profile, the use of this database does not seem to have implications in relation to the general scenario in which the indicators proposed here were tested.

## 5. Conclusions

This study offers simple and valid summary indicators for characterizing the consumer food environment in establishments that sell foods for immediate consumption. The *a priori* classification of types of establishments and the proposed indicators displayed good discriminatory power in the context studied and are easy to apply, interpret, and adapt to different contexts. The use of these indicators in studies and public policies and in the establishments’ routine management will allow a concise and diversified diagnosis of the availability of foods, expressing the degree to which these environments favor or hinder healthy food choices.

## Figures and Tables

**Figure 1 foods-10-01434-f001:**
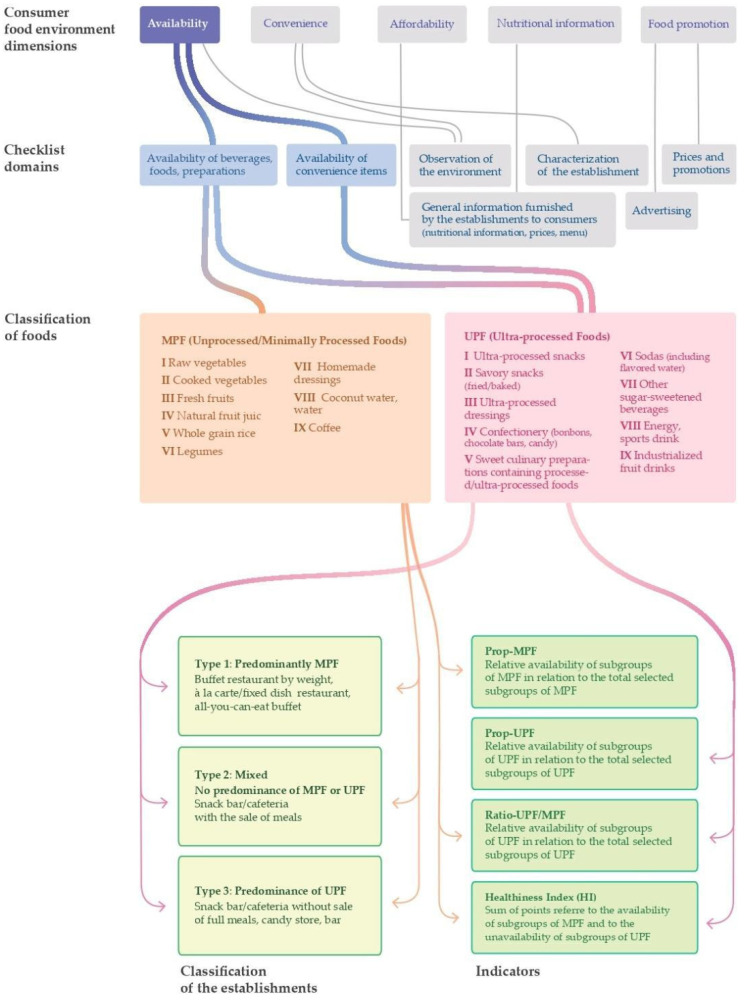
Flowchart for the development of healthiness indicators for commercial establishments that sell foods for immediate consumption.

**Table 1 foods-10-01434-t001:** Indicators for assessment of availability of foods in establishments that sell foods for immediate consumption *.

Indicator (Acronym)	Formula for Calculation	Purpose	Example
Proportion of availability of subgroups of unprocessed/minimally processed foods (MPF ^1^) among all the selected subgroups of MPF (Prop-MPF)	(Number of subgroups of MPF sold/total selected subgroups of MPF) × 100	Expresses the relative availability of subgroups of MPF in relation to total selected subgroups of MPF	Situation: the establishment sells five of the nine selected subgroups of MPF (5/9) × 100 = 55.5% Interpretation: the establishment sells more than half (55.5%) of the selected subgroups of MPF.
Proportion of availability of subgroups of ultra-processed foods (UPF ^2^) among all the selected subgroups of UPF (Prop-UPF)	(Number of subgroups of UPF sold/total selected subgroups of UPF) × 100	Expresses the relative availability of subgroups of UPF in relation to total selected subgroups of UPF	Situation: the establishment sells eight of the nine selected subgroups of UPF. (8/9) × 100 = 88.9% Interpretation: the establishment sells the vast majority (88.9%) of the selected subgroups of UPF.
Ratio between availability of UPF and availability of MPF (Ratio-UPF/MPF)	Number of subgroups of UPF sold/number of subgroups of MPF sold	Identifies the relative advantage/disadvantage in the availability of MPF in relation to that of UPF. Allows assessing the “competition” between the groups of foods.	Situation: the establishment sells eight of the nine selected subgroups of UPF and five of the nine selected subgroups of MPF 8/5 = 1.60 Interpretation: the establishment sells 1.6 times more subgroups of UPF than subgroups of MPF, representing 60% greater availability of subgroups of UPF compared to subgroups of MPF.
Healthiness Index (HI)	Total availability of MPF subgroups and unavailability of UPF subgroups: One point is assigned for each subgroup of MPF sold One point is assigned for each subgroup of UPF not sold No points are assigned for each subgroup of MPF not sold No points are assigned for each subgroup of UPF sold (Total score of subgroups of MPF sold + total score of subgroups of UPF not sold)/18 × 100	Indicates with a summary measure the establishment’s healthiness based on assessment of the availability of subgroups of MPF and unavailability of subgroups of UPF, varying from zero to 100. The closer the score is to 100, the healthier the establishment. Expresses the idea that an establishment that promotes healthy eating both facilitates access to MPF and hinders access to UPF.	Situation: The establishment sells: 5 subgroups of MPF (5 points) 8 subgroups of UPF (0 points) The establishment does not sell: 4 subgroups of MPF (0 points) 1 subgroup of UPF (1 point) HI = [(5 + 1)/18] × 100 HI = 33.43 Interpretation: the establishment reached 1/3 of the maximum score (100).

^1^ MPF: unprocessed/minimally processed foods: (i) raw vegetables, (ii) cooked vegetables, (iii) fresh fruits, (iv) natural fruit juice, (v) whole grain rice, (vi) legumes, (vii) homemade dressings, (viii) coconut water and water, and (ix) coffee. ^2^ UPF: ultra-processed foods: (i) ultra-processed snacks (cereal bars; breakfast cereal; whole grain cookies; cream-filled cookies; cookies without filling; packaged savory snacks or crackers without filling), (ii) savory snacks (fried/baked), (iii) ultra-processed sauces (readymade dressings; shoyu), (iv) confectionery (chocolate bonbons and chocolate bars; candies), (v) sweet culinary preparations containing processed/ultra-processed foods (eclairs, cupcakes, cakes, pies), (vi) sodas (including flavored water), (vii) other sugar-sweetened beverages (ready-to-drink tea; guarana drink; industrialized fruit juices; soy-based industrialized beverages; flavored milk or milk-based bevchart erages or industrialized yogurt), (viii) energy and sports drinks; and (ix) industrialized fruit drinks. *In the present study, 18 items, as listed in (1) and (2), were selected to compose the indicators. The same indicators formulas can be applied to instruments with different numbers of foods.

**Table 2 foods-10-01434-t002:** Application of indicators of availability of foods for immediate consumption in commercial establishments located in two public universities, according to types of establishments. Rio de Janeiro, Brazil, 2015–2016.

Types of Establishments	Prop-MPF ^4 ^ % Mean (CI) ^8^	Prop-UPF ^5 ^ % Mean (CI)	Ratio-UPF/MPF ^6 ^ Mean (CI)	HI ^7^ Mean % (CI)
Total (n = 78)	45.16 (39.63–50.68)	71.51 (66.47–76.55)	2.46 (1.97–2.96)	36.82 (32.90–40.74)
Type 1 (n = 14) ^1^	61.91 (53.67–70.14)	42.86 (31.37–54.35)	0.72 (0.52–0.93)	59.52 (53.06–65.99)
Type 2 (n = 30) ^2^	58.15 (50.24–66.06)	78.52 (72.66–84.38)	1.57 (1.26–1.88)	39.82 (35.56–44.07)
Type 3 (n = 34) ^3^	26.80 (20.68–32.92)	77.12 (70.45–83.80)	4.02 (3.16–4.88)	24.84 (20.33–29.35)

^1^ Type 1: Establishments that predominantly sell unprocessed/minimally processed foods (MPF). ^2^ Type 2: Establishments in which there is no predominance of the availability of MPF versus ultra-processed foods (UPF). ^3^ Type 3: Establishments that predominantly sell UPF. ^4^ Prop-MPF: Relative availability of subgroups of MPF in relation to the total selected subgroups of MPF. ^5^ Prop-UPF: Relative availability of subgroups of UPF in relation to the total selected subgroups of UPF. ^6^ Ratio-UPF/MPF: Ratio between the availability of UPF and the availability of MPF. ^7^ HI: Healthiness Index. ^8^ CI: 95% confidence interval.

**Table 3 foods-10-01434-t003:** Application of *a priori* classification of establishments and indicators of availability of foods for immediate consumption in establishments in two public universities, Rio de Janeiro, Brazil, 2015–2016.

Indicators	UERJ (*n* = 25)	UFRJ (*n* = 53)
Classification of establishments (%)		
Type 1 ^1^	4.0	24.5
Type 2 ^2^	36.0	39.6
Type 3 ^3^	60.0	35.8
Prop-MPF ^4^		
Mean (CI) ^5^	33.8 (23.2–44.3)	50.5 (44.4–56.7)
Min-Max	0.00–88.9	11.1–100.0
Median	22.2	55.6
Interquartile range	44.4	33.3
Prop-UPF ^6^		
Mean (CI)	77.8 (69.1–86.5)	68.6 (62.4–74.8)
Min-Max	11.1–100.0	11.1–100.0
Median	88.9	66.7
Interquartile range	22.2	33.3
Ratio-UPF/MPF ^7^		
Mean (CI)	3.8 (2.7–5.0)	1.9 (1.4–2.3)
Min-Max	0.2–8.0	0.2–8.0
Median	3.3	1.3
Interquartile range	5.3	1.6
HI ^8^		
Mean	28.0 (21.0–35.0)	41.0 (36.5–45.4)
Min-Max	11.1–77.8	11.1–77.8
Median	22.2	44.4
Interquartile range	19.4	25.0

^1^ Type 1: Establishments that predominantly sell unprocessed/minimally processed foods (MPF). ^2^ Type 2: Establishments in which there is no predominance of the availability of MPF versus ultra-processed foods (UPF). ^3^ Type 3: Establishments that predominantly sell UPF. ^4^ Prop-MPF: Relative availability of subgroups of MPF in relation to total selected subgroups of MPF. ^5^ CI: 95% confidence interval. ^6^ Prop-UPF: Relative availability of subgroups of UPF in relation to total selected subgroups of UPF. ^7^ Ratio-UPF/MPF: Ratio between the availability of UPF and availability of MPF. ^8^ HI: Healthiness Index.

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
