# Peer review of "Development and Application of Healthiness Indicators for Commercial Establishments That Sell Foods for Immediate Consumption"

_foods, 2021, doi:10.3390/foods10061434_

Round 1

Reviewer 1 Report

None.

Author Response

Thanks for the review.

Reviewer 2 Report

This research is interesting for academics and professionals. I think the authors have done a great effort to provide indicator for classifying establishments that sell foods for immediate consumption. I have several comments and suggestions to improve this research:

  • The paper suffers from important mistakes in terms of format. For example, in the Introduction section, the first paragraph corresponds to the guide for authors for this section.
  • The theoretical background of the manuscript is brief and does not explain and justify under an academic perspective the research objectives, especially in the field of commercial establishments.
  • The data collection was conducted from November 2015 to February 2016, around 5 years ago. The authors shoud justify to what extent this can affect the results of the research.
  • The academic discussion of the results is suitable. However, I think the managerial implications of the research should be explained in depth.

Round 2

Reviewer 2 Report

In my opinion, the Introduction and Implications should be reinforced. However, I do not see a positive attitude of the authors to substantially improve the manuscript. For example, the changes in the managerial implications were minor and not based on an managerial discussion of the results. Additionally, the theoretical justification of the gap in the literature and the research objectives is not enough.

Author Response

This manuscript is a resubmission of an earlier submission. The following is a list of the peer review reports and author responses from that submission.

Round 1

Reviewer 1 Report

Review of the manuscript: “Development and application of healthiness indicators for commercial establishments that sell foods for immediate consumption”

The Authors developed and implemented an a priori classification of food environments that sell food for immediate consumption, and a relevant set of indicators of food availability. The application of the indicators in two Brazilian Universities revealed the good discriminatory power, the ease of application and flexibility of the method.

This paper is thoughtfully constructed, the rationale for the work is compelling and the methods and analysis are sound.

The writeup is friendly to follow and the theoretical thinking and estimations are coherent. Results and conclusions are in line with the empirical parts of the study. The discussion on the strengths, transferability and limits of the proposed indicators make sense as well.   

Few minor comments and reviews are suggested below:

  1. I would mention the extended wording of the acronyms MPF and UPF, both in the abstract and when they are mentioned for the first time in the text. E.g., in the Abstract: substitute “based on the availability of MPF (unprocessed, minimally processed, or processed foods and culinary preparations based on these foods) and UPF (ultra-processed foods and culinary preparations containing these foods) in the establishments” with “based on the availability of Unprocessed/Minimally Processed Foods (MPF) and Ultra-processed Foods (UPF) in the establishments”. The detailed description is redundant in an abstract and do not clarify the acronyms.
  2. I suggest adding a graphic scheme to the methodological section, in order to visually clarify the relationships among the dimensions of the checklist, the relevant domains, food and establishment classifications and indicators. A sort of graphic workflow, or diagram of methodological approach, would facilitate the comprehension of the work rationale.
  3. It seems to me that section 3.1 is part of the methodology. I would move this section, and unify it with section 2.4.
  4. Chart 1. In the cell of the Purpose of the Healthiness index check the spelling of “varying from zero do 100”
  5. Line 212-213: even if the assumption is ascertain and well-known, I would add some references.
  6. Line 219-221: Delete the sentence “This section may be divided… that can be drawn”.
  7. Line 224: Do you mean “group 2” or “type 2”? Which group are you referring to?

Reviewer 2 Report

This manuscript is focused on a topic that is very interesting to marketing academics and managers. However, in my opinion, this research needs substantial improvements in terms of theoretical foundations, conceptual framework, methodology and contributions to research field.

The Introduction section is very brief and weak. It does not provide a significant theoretical framework and the justification of the objectives of the research. This section has to be substantially extended and improved. What has been done previously in this field? (This part is more developed in the discussion section than in the introduction). Which have been the main indications provided in the literature? How can we define those indicators that which are their main characteristics? Are they only related to healthiness? What is the relevance of the elaboration of a summary of indications for characterizing food establishments?

The authors should justify in the theoretical framework of the manuscript why it is of special interest the development of indicators for establishments in public universities and not other type of places or establishments.

Why the instrument provided by Glanz et al. (2005) and Caspi et al. (2012) is preferable than others? This needs further discussion.

The authors assert: "our interest was to develop indicators based on the availability of foods". Why this dimension and not others?

Regarding the methodology, the authors should explain how this method responds to the research objective. Moreover, the main characteristics of the audit method must be explained.

Regarding the data collection, the main information is missing or not clear. How was the sample procedure? Which is the sample unit? How the participants were selected? Where is the questionnaire or the research instrument? What is the representativeness of the results in those universities?

In the factor analysis, two factors were fixed. Did the authors develop additional analyses in order to figure out the number of factors without any previous condition? Which was the rotation in this analysis? Was it exploratory or confirmatory?  More information must be provided in this section.

The managerial implications of this research are missing. I am not sure about how this research can contribute to marketing management. The authors need to clarify this.

Finally, I think in general the paper is well structured. However, I think it needs a proofreading in order to improve the clarity of expression and readability, together with the correction of some mechanical, grammatical or organizational errors.

Best wishes in the refining process of the current study.